# Efficient Redirection of NK Cells by Genetic Modification with Chemokine Receptors CCR4 and CCR2B

**DOI:** 10.3390/ijms24043129

**Published:** 2023-02-04

**Authors:** Frederik Fabian Feigl, Anika Stahringer, Matthias Peindl, Gudrun Dandekar, Ulrike Koehl, Stephan Fricke, Dominik Schmiedel

**Affiliations:** 1Fraunhofer Institute for Cell Therapy and Immunology (IZI), 04103 Leipzig, Germany; 2Chair of Tissue Engineering and Regenerative Medicine (TERM), University Hospital of Würzburg, 97070 Würzburg, Germany; 3Translational Center Regenerative Therapies, Fraunhofer Institute for Silicate Research (ISC), 97070 Würzburg, Germany; 4Institute for Clinical Immunology, University of Leipzig, 04103 Leipzig, Germany

**Keywords:** chemokine receptor, migration, immune cell infiltration, trafficking, NK cells, immunotherapy, CCR2, CCR4, genetic engineering

## Abstract

Natural killer (NK) cells are a subset of lymphocytes that offer great potential for cancer immunotherapy due to their natural anti-tumor activity and the possibility to safely transplant cells from healthy donors to patients in a clinical setting. However, the efficacy of cell-based immunotherapies using both T and NK cells is often limited by a poor infiltration of immune cells into solid tumors. Importantly, regulatory immune cell subsets are frequently recruited to tumor sites. In this study, we overexpressed two chemokine receptors, CCR4 and CCR2B, that are naturally found on T regulatory cells and tumor-resident monocytes, respectively, on NK cells. Using the NK cell line NK-92 as well as primary NK cells from peripheral blood, we show that genetically engineered NK cells can be efficiently redirected using chemokine receptors from different immune cell lineages and migrate towards chemokines such as CCL22 or CCL2, without impairing the natural effector functions. This approach has the potential to enhance the therapeutic effect of immunotherapies in solid tumors by directing genetically engineered donor NK cells to tumor sites. As a future therapeutic option, the natural anti-tumor activity of NK cells at the tumor sites can be increased by co-expression of chemokine receptors with chimeric antigen receptors (CAR) or T cell receptors (TCR) on NK cells can be performed in the future.

## 1. Introduction

Solid tumors are considered as complex tissues that develop through genetic and epigenetic changes in tumor cells and restructuring of the tumor microenvironment (TME) [1,2,3]. The TME is a highly heterogeneous milieu consisting of tumor cells, fibroblasts, endothelial cells, immune cells and non-cellular components such as collagen, fibronectin, hyaluronan, laminin and others [1,2,3]. The cellular composition and spatial localization of the different cells within the TME is guided by chemokines secreted by tumor cells, stromal cells and immune cells. The infiltration of the TME by immune cells is considered a key factor in cancer prognosis for many tumor types [4,5,6,7]. Activating immune cells such as cytotoxic T cells (CTLs) or Natural Killer (NK) cells has predominantly anti-tumor effects and is associated with overall improved clinical outcomes, e.g., by determining responsiveness to checkpoint inhibitor therapies when found within the tumor [8,9,10,11]. However, immunosuppressive immune cells such as regulatory T cells (T regs), myeloid-derived suppressor cells (MDSC), tumor-associated macrophages (TAM) and tumor-associated neutrophils (TAN) act in a pro-tumorigenic way and efficiently suppress the anti-tumor immune responses by diverse means [8,12,13]. Next, to direct effects of chemokines on tumor cell motility and their ability to metastasize, tumors recruit immune-suppressive cell populations by means of these chemokines and thereby also benefit from facilitated immune escape. Consequently, chemokines that recruit T regs, monocytes or macrophages, such as CCL22 [14], CCL17 [12,15], CCL2 [16] and others, are often secreted in high quantities in the TME, whereas infiltration of activating immune cells is frequently poor due to a lack of the respective chemokines [16,17]. Especially CCR4 expression in the tumor is correlated with failure of immunotherapies [18] and a target for therapeutic intervention, e.g., by agonists, antibody therapy or chimeric antigen receptors (CAR-) T cell immunotherapy [19,20,21,22,23].

Unmodified, blood-borne, cytokine-activated NK cells have been used for adoptive transfer in patients suffering from melanoma, renal cell carcinoma and other cancer types for many years [24]. NK cells are cytotoxic lymphocytes that can recognize transformed cells by means of a vast array of germline-encoded activating and inhibitory receptors, and are capable of killing tumor cells without prior sensitization [24,25]. Both patient-derived (autologous) NK cells and NK cells from healthy donors (allogeneic) can be safely used for cancer therapy, as an HLA-mismatch between donor and recipient does not induce graft-versus-host disease (GvHD) [24,26]. Additionally, an NK cell line, called NK-92, maintains cytotoxic capabilities towards tumor cells and is under clinical investigation [26,27]. NK-92 cells express most activating NK-cell receptors but lack most of the inhibitory killer cell immunoglobulin-like receptors (KIRs) [27].

To enhance the cytotoxic potential and redirect NK cells more efficiently towards tumor cells, NK cells have been previously equipped with CARs [28,29]. However, while the treatment of several malignancies of hematopoietic origin [30,31,32,33,34] yielded promising results, NK cell-based immunotherapies in solid tumors require further improvements beyond a CAR modification to successfully treat these cancer types [35,36,37,38,39,40]. One profound explanation of treatment failure is the poor infiltration into the tumor sites [39,41,42,43]. 

To alter the migration of NK cells, some studies overexpressed the NK cell chemokine receptor CXCR4 or variants thereof on NK cells to enhance migration to the bone marrow [44,45]. Expression of CCR7, a receptor found on subsets of primary NK cells [46], on NK-92 cells redirected the cells towards CCL19-expressing lymphoma cells and enhanced tumor clearance in xenograft models [47]. Additionally, an overexpression of CCR5, which is naturally relevant to recruit NK cells to infected lung tissues [48,49], was successfully used in a combination therapy with CCL5-armed oncolytic viruses to enhance their antitumor activity [50]. Another study showed that the restored expression of CXCR2, which is lost on NK cells during in vitro cultivation and in the blood of tumor patients, can enhance their infiltration [51]. These studies underline the stunning potential to enhance the efficacy of cell therapies by engineering immune cells with chemokine receptors. 

However, whether receptors that are typically not found on the NK cell lineage, such as CCR2B or CCR4, can be expressed on NK cells and redirect them towards CCL2 or CCL17/22, respectively, has not been investigated before. These further modifications can potentially improve the anti-tumor properties of NK cell therapies, especially for the treatment of solid malignancies.

## 2. Results

### 2.1. CCR4 and CCR2B Can Be Efficiently Expressed on NK-92 Cells

The chemokine receptors CCR4 and CCR2B were cloned from human peripheral mononuclear cells into a lentiviral expression vector. Thereby, the CCR2 isoform CCR2B was isolated and used throughout all experiments, as CCR2B is the predominant isoform expressed on monocytes [52,53]. 

To assess, if NK cells can be modified with these chemokine receptors of different immune cell lineages, we first used the NK cell line NK-92 as a model and transduced them with baboon endogenous virus (BaEV)-enveloped lentiviral particles encoding CCR4 or CCR2B. Naturally, NK-92 cells express neither CCR4 nor CCR2 (Figure 1A,D). However, following transduction and selection, more than 80% of the cells transduced with CCR4 showed expression of the receptor. For CCR2B, around 70% of the selected NK-92 cells showed expression of the receptor (Figure 1A,D). While transgene expression of CCR4 was constant over four weeks after selection, CCR2B expression was lost by 30% of the cells during that time. To compare the expression of transduced NK-92 cells with native expression levels of the receptors, T regs and monocytes were stained with specific antibodies against CCR4 and CCR2B, respectively. For both receptors, fluorescence intensities of CCR4-engineered NK-92 cells increased 2.5-fold compared to T regs and CCR2B expression was increased 3-fold compared to monocytes. To assess mRNA levels of these receptors, cDNA was generated and qPCR was performed, using *β*-actin as reference gene. At the mRNA level, CCR4 expression was increased 2.5-fold compared to T regs and CCR2B expression was increased 3-fold compared to monocytes (Figure 1B,E), in line with the differences observed for receptor surface expression. Therefore, the translation and trafficking of CCR4 and CCR2B receptors seems to be equally efficient in NK-92 cells as in T regs and monocytes. Immunofluorescence microscopy confirmed that the chemokine receptors are evenly distributed on the surface of the NK-92 cells (Figure 1C,F). 

### 2.2. Genetically Engineered NK-92 Cells Efficiently Migrate towards CCL22 or CCL2

The chemotactic activity of the genetically modified cells was examined using a Boyden chamber migration assay [54]. Chemotaxis of the NK-92 cells from the upper well towards recombinant chemokines in the lower well was examined after 2 h by counting the absolute number of migrated cells. For CCR4-transduced NK-92 cells, the chemokine CCL22 was used as the chemotactic factor. CCR2B-expressing NK-92 cells were attracted by their ligand CCL2. The number of migrated cells was significantly increased in both transfectants when the corresponding ligand was added to the lower chamber. Both CCR4 transduced cells and CCR2B transduced cells showed an approximately 10-fold increase compared to untransduced cells or to non-specific migration towards pure medium. As expected, no significant chemotactic activity was observed in the non-transduced cells, neither towards CCL22 nor towards CCL2 (Figure 2A). As a positive control for migration, T regs and monocytes were analyzed to compare their chemotactic activity with the modified NK-92 cells. Within this experimental setup, T regs showed an approximately 2-fold enhancement of migration, while monocytes showed an approximately 7-fold enhancement of migration towards the cognate ligand compared to non-specific migration when no chemokine was added to the medium in the lower well (Figure 2B). To confirm the findings observed for recombinant chemokines, we generated HEK293T cells overexpressing CCL22 or CCL2, respectively. In line with the data for the recombinant chemokines, NK-92 cells carrying the respective receptors efficiently migrated towards the ligand-secreting HEK293T cells (Figure 2C). The data indicate that the CCR4 and CCR2B receptors can efficiently transduce signals in NK-92 cells and induce cell migration towards their respective ligands.

### 2.3. CCR4 or CCR2B Expression Does Not Impact the Cytotoxic Activity of NK-92 Cells

Next, we tested if the transduced receptor may have stimulatory or adverse effects on NK-92 cell receptor expression or cytotoxicity. To this end, we incubated CCL2 with CCR2B-NK-92 and control cells, as well as CCL22 with CCR4-NK-92 and control cells, for 16 h and then analyzed an array of NK cell receptors (Figure 3A). We did not observe any changes in NK cell receptor expression analyzed by flow cytometry, independent of receptor expression and presence of the respective ligand in the medium. Consequently, we did not observe significant changes in cytolytic activity of either CCR4- or CCR2B-NK-92 cells compared to parental NK-92 when being co-cultured directly with HEK293T cells overexpressing CCL22 or CCL2 chemokines (Figure 3B).

Therefore, we concluded that the overexpression of CCR4 and CCR2B selectively changes migratory behavior of NK-92 cells without affecting their ability to lyse tumor cells.

### 2.4. CCR4 and CCR2B Expression and Functional Testing on Peripheral Blood NK Cells

Although NK-92 cells are under clinical investigation and FDA-approved, we investigated whether the genetic modification with these two chemokine receptors could also be applied to peripheral blood NK cells, as these have superior properties for later clinical application [31]. Neither freshly isolated primary (day 0) nor cytokine-activated primary NK cells (day 7) expressed CCR4 or CCR2B (Figure 4A). For the transduction, we used a *γ*-retroviral instead of a lentiviral system for the gene delivery of the chemokine receptors into the IL-2 and IL-15 activated peripheral blood NK cells. Viral supernatants were used either fresh or from frozen stocks for the different NK cell donors. Transduction efficiencies are shown in Figure 4B. A high donor variability regarding the chemokine receptor-bearing population ranging from 10–45% for CCR4 and 10–50% for CCR2B was observed. For CCR2B, untransduced NK cells showed some background signal up to 5% of the total population. The MFIs of each positive population were rather constant for CCR4, for CCR2B a wider deviation of the values was displayed, but on a higher level (Figure 4B). Migration experiments were carried out after each round of transduction; however, only for transduction efficiencies greater than 15% could significant migration behavior be observed. The migration effect was not as pronounced as in NK-92 cells, which might result from the lower fraction of CCR4/CCR2B positive cells. Nevertheless, we could detect specific enhanced migration towards the ligands CCL22 and CCL2 compared to untransduced control cells (Figure 4C). To investigate the stability of the transgene expression, cells were checked for surface expression of CCR4 or CCR2B 3, 7, 10 and 14 days after transduction. Transgene expression was stable for both constructs. The fraction of cells positive for the respective chemokine receptor remained constant or even slightly increased over time, as observed for both constructs (Figure 4D).

### 2.5. Characterization of NK Cell Populations between Transduced and Untransduced Fractions

CCR4 or CCR2B transduced cells were further characterized and compared to the untransduced fraction within one transduction experiment to determine the phenotype of NK cell populations susceptible for transduction. First, we analyzed the expression of the surface markers CD56 and CD16 and compared the ratios of CD16^+^/CD16^−^ NK cells within transduced and untransduced NK cells (Figure 5A). The CCR-transduced fractions showed significantly higher ratios, suggesting that more transduced cells express CD16 compared to the untransduced fraction. We further analyzed the expression of the maturation marker CD57. The population of CD57-positive cells was significantly smaller within transduced cells with differences ranging up to 5%. We also looked for surface expression of KIR2D. Again, we found significant differences, with more KIR2D expressing NK cells within the transduced fraction. Furthermore, we were analyzing cells expressing either the activation-induced activating receptor NKp44 or the inhibitory receptor NKG2A. In both cases, we could not detect any significant differences between transduced and untransduced cells (Figure 5B). Last, cells were stained with CellTrace™ Violet prior to transduction to track proliferation of the cells (Figure 5C). After four days of culture in presence of IL-2 and IL-15, the CellTrace™ Violet signal of the transduced cells was less intense compared to untransduced cells, meaning that the transduced cells tend to proliferate more, which matches the observations in Figure 4D that the frequency of CCR-positive cells was slightly increasing over time.

## 3. Discussion

NK cells possess great potential to become an “off-the-shelf” cell therapy product against cancer as they can be safely used for allogeneic cell transfers with low or no occurrence of GvHD [55]. However, whereas preclinical results for hematological malignancies are promising [56,57,58,59], for solid tumors, major hurdles need to be overcome, as the immune cells frequently do not reach the tumor sites, or, if they do, quickly exhaust in the immunosuppressive TME [60,61,62]. Here, we show a potential approach to enhance NK cell trafficking towards solid tumor sites via the overexpression of natural chemokine receptors expressed on tumor promoting immune cells that are often detected in solid tumor tissues.

The chemokine receptors CCR4 and CCR2B, which are naturally found, amongst other cell types, on T regs and monocytes, respectively [17], were selected to be overexpressed in the NK cell lineage. Neither receptor is known to be expressed on NK-92 cells [63] and in accordance we did not detect expression of CCR4 or CCR2 prior to transduction, nor migration towards their respective ligands. Regarding the expression of CCR4 and CCR2 on primary NK cells, published studies are inconsistent [63,64]. While in one study neither CCR4 nor CCR2 was detected [64], in another both receptors could be detected to some extent on NK cells [65]. We could neither detect the receptors on freshly isolated nor on IL-2 and IL-15 activated NK cells from peripheral blood of healthy donors; therefore, it is likely that NK cells used in immunotherapeutic approaches are devoid of both receptors.

CCR4 and CCR2B seem particularly well suited for overexpression in cell therapy products. Both receptors largely contribute to immune escape [66] and resistance to immunotherapy [18], and have been suggested as tumor-associated targets for therapy, as they may be expressed by tumor cells as well [19,67]. The immune escape factors CCL22/CCL17 and CCL2 are particularly interesting chemokines to recruit activating immune cells by genetic engineering, as the selective pressure posed by CCR4/CCR2B-NK cells on tumor cells can be very advantageous: a reduction of these chemokines in response to this treatment in the TME would compromise the recruitment of immunosuppressive immune cells as well. As a consequence, immune escape from the patient’s immune system would be impaired and other therapeutic interventions may be enabled.

One major drawback of this study is the limitation of in vitro data. The recombinant ligands CCL22 and CCL2 in this study were used in concentrations of 50 ng/mL, and it is not clear from our data what concentration will be required in vivo to efficiently reroute immune cells towards tumor sites. It is likely that the applied concentrations do not reflect the physiological concentrations at tumor sites. CCL22, for example, is only present at concentrations up to 1 ng/mL in ascites and tissue samples of ovarian cancer [68], as are CCL2 levels in the plasma of breast cancer patients [69]. Then again, the expression of both receptors on NK-92 cells was slightly above physiological levels and the migration in the experiments using cell culture inserts was high, with about 20% of cells migrating towards the chemokine within two hours. This indicates a very efficient migratory behavior of genetically engineered cells comparable to cells naturally expressing these receptors. Additionally, by combining a mesothelin-specific CAR with CCR2B, CAR-T cells showed high functionality and improved efficacy in a solid cancer model, suggesting that genetically added chemokine receptors can also improve cell therapies in vivo [70].

One major disadvantage of the NK-92 cell line is its origin from a lymphoma which causes the necessity to irradiate the cell prior to infusion into the patient, resulting in limited persistence and lower cytotoxic functions [71]. Our study shows that the approach of genetically modifying the cells with chemokine receptors is also applicable to peripheral blood NK cells, which makes it more attractive for future immunotherapies. Our data imply that the cytotoxic activity of the modified cells is not altered. Furthermore, we could show that the transgene expression in peripheral blood NK cells is kept stable over at least 14 days, which means prolonged migration capacity after infusion. We further found out that among the transduced cells, the population bearing CD16 on its surface is increased compared to the untransduced fraction, which could lead to higher cytotoxicity induced by CD16 via antibody dependent cell-mediated cytotoxicity (ADCC) [72]. On contrary, fewer CD57 expressing cells were detected in the transduced fraction. As CD57 is regarded as a maturation marker, only expressed on fully differentiated NK cells, its expression is expected to be correlated with the expression of CD16 [73,74]. However, our study showed a preference of the transduced cells to express CD16 but not CD57. This CD57−CD16+ subset is thought to proliferate more when stimulated with cytokines, but possesses a lower cytotoxic activity compared to the CD57+ CD16+ subset [74,75]. The differences regarding the expression of KIR2D indicate a mature phenotype of the transduced cells [76]. Taken together, our results propose that NK cells transduced with the chemokine receptors tend to proliferate more and express more CD16 and KIR2D, while expressing less CD57. Such a CD16+/KIR2D+/CD57- population might not yet be finally differentiated, but already exhibits a mature phenotype with high cytotoxic activity and proliferation potential. This is also in accordance with a study in which it was shown that functionally mature but not terminally differentiated NK cells represent the main population to be transduced by BaEV-encoded lentiviral particles [77]. Therefore, the incorporation of chemokine receptors together with CARs or T cell receptors (TCRs) that have shown promising results against several types of cancer might be a promising approach, as solid tumor infiltration is still a major hurdle in immunotherapy [17,78]. For NK cells, one study showed the overexpression of CXCR4, a chemokine receptor commonly found on NK cells, together with a CD19-specific CAR [45], raising hopes that such double gene-edit may also yield enhanced efficacy of cell therapies in other cancer types.

Our study shows that NK cells could be modified with a variety of different chemokine receptors that mediate efficient migration towards their ligands. A measurement of an array of cancer-associated chemokines may not only provide insights into intratumoral immunity but also dictate the choices for cell therapy engineering. In the future, NK cell-based immunotherapies may be further personalized by equipping NK cells with chemokine receptors in dependence of the intratumoral chemokine secretion patterns.

## 4. Materials and Methods

### 4.1. Cell Lines and Primary Cells

All cells were grown at 37 °C with 5% CO_2_ and split every 2 to 3 days. HEK293T cells (DSMZ, Braunschweig, Germany) were grown in DMEM (Sigma-Aldrich, St. Louis, MO, USA) with 10% fetal calf serum (FCS) (Sigma-Aldrich, St. Louis, MO, USA). NK-92 cells (DSMZ, Braunschweig, Germany) were cultured in RPMI1640 (Thermo Fisher, Waltham, MA, USA), supplemented with 20% FCS (Sigma-Aldrich, St. Louis, MO, USA) and 500 U/mL IL-2 (Peprotech, Cranbury, NJ, USA).

PBMCs were isolated from buffy coats of five healthy donors bought from the Institute for Transfusion Medicine of the University Clinic in Leipzig. The buffy coat was diluted 1:2 with PBS and centrifugation was carried out at 400× *g* for 30 min using a Ficoll-Paque (Cytiva, Marlborough, MA, USA) gradient. The PBMC fraction was transferred in a sterile 50 mL tube with a 10 mL serological pipette. Primary NK cells were then purified by negative selection using NK cell isolation kit according to the manufacturer’s instructions (MojoSort NK cell isolation Kit, Biolegend, San Diego, CA, USA). After isolation, 0.5 × 10^6^ cells were cultured per well in a 24-well plate in 500 µL NK Medium (NK MACS Medium (Miltenyi Biotec, Bergisch Gladbach, Germany), 5% Human Serum (Miltenyi Biotec, Bergisch Gladbach, Germany), 500 U/mL IL-2 and 140 U/mL IL-15 (Peprotech, Cranbury, NJ, USA)). The purity of NK cells after isolation was determined by flow cytometry. All NK cells that were used had a purity greater than 80% on the day of isolation.

Isolated T regs were kindly provided as frozen aliquots from a single donor by Dr. Anna Dünkel (Fraunhofer IZI, Leipzig, Germany). Monocytes were freshly isolated from three different donors and kindly provided by Dr. Claire Fabian (Fraunhofer IZI, Leipzig, Germany).

### 4.2. Transgene Constructs

The sequences of the chemokine receptors CCR2 and CCR4 and their ligands CCL2 and CCL22 were amplified from PMBC cDNA using following primers (Thermo Fisher, Waltham, MA, USA): CCR2_fw (ATTCTAGAGCCGCCACCATGCTGTCCACATCTCGT), CCR2_VarB_rev (ATGAATTCGAGACGTTATAAACCAGCCGAGACTTCC), CCR4_fw (ATTCTAGAGCCGCCACCATGAA-CCCCACGGATATAG), CCR4_rev (ACTATGAATTCCTACAGAGCATCATGGAGAT), CCL2_fw (ATTCTAGAGCCGCCACCATGAAAGTCTCTGCCGCC), CCL2_rev (ACTATGAATTCTCAAGTCTTCGGAGTTTGGG), CCL22_fw (ATTCTAGAG-CCGCCACCATGGATCGCCTACAGACT), CCL22_rev (ACTATGAATTCTCATTGGCTCAGCTTATTGAG)

PCR products were purified using a Zymoclean Gel DNA Recovery kit according to the manufacturer’s instructions (Zymo Research, Irvine, CA, USA). CCR4/CCR2B sequences were cloned in the lentiviral expression vector hEF1α-H2B-mVenus-IRES-mCherry-PGK-Puromycin (Addgene #99278 [79], kindly provided by Jordan Miller). Ligations were performed with 50 ng vector and a molar insert to vector ratio of 3:1 using T4 DNA Ligase according to the manufacturer’s instructions overnight at 16 °C (NEB). The whole ligation mixture (20 µL) was then added to 20 µL of chemical competent *E. coli* TOP10 (Thermo Fisher, Waltham, MA, USA) and incubated on ice for 30 min followed by a 30 s heat shock at 42 °C and 5 min on ice. Then, 150 µL of SOC medium was added and incubated at 37 °C, 750 rpm for 1 h. Afterwards, 50 µL were plated on LB agar plates containing 50 µg/mL Kanamycin. Colonies were grown overnight at 37 °C. Plasmids were isolated using the ZymoPURE MiniPrep Kit according to the manufacturer’s instructions (Zymo Research, Irvine, CA, USA).

### 4.3. γ-Retroviral Vector Generation

*γ*-Retroviral vector particles were generated by transient transfection of HEK293T cells using TransIT-VirusGEN (MirusBio, Madison, WI, USA) and third generation lentiviral plasmids (hEF1*α*-H2B-mVenus-IRES-mCherry PGK-Puromycin vector (Addgene plasmid #99278 [79]) containing DNA sequences for CCR4, CCR2, CCL22 or CCL2; BaEVRless envelope [80] protein in pTwist CMV BetaGlobin WPRE Neo vector (Twist Bioscience, South San Francisco, CA, USA), pMDLg/pRRE-gagpol and pRSV-Rev (Addgene #12251 [81] and #12253 [81], kindly provided by Didier Trono)). One day before transfection, 1 × 10^5^ HEK293T cells were seeded in each well of a 12-well plate. In total, 1 µg DNA was added to 100 µL DMEM without supplements, together with 3 µL TransIT-VirusGen. *γ*-retroviral vector particles were generated by transient transfection of HEK293T cells using TransIT-VirusGEN (MirusBio) and *γ*-retroviral plasmids (pBullet vector [82] (kindly provided by Reno Debets, Erasmus University Medical Center) containing DNA sequences for CCR4 or CCR2; pHIT60-gagpol [83] and BaEVRless envelope [80] protein in pTwist CMV BetaGlobin WPRE Neo vector (Twist Bioscience, South San Francisco, CA, USA)). One day before transfection, 1.5 × 10^5^ HEK293T cells were seeded in each well of a 6-well plate. In total, 1.9 µg DNA was added to 200 µL DMEM without supplements, together with 5.8 µL TransIT-VirusGen. The transfection solution was mixed and incubated for 15 min at room temperature before it was added dropwise to the cells. After two days, the cell culture supernatant was collected and filtered through a 0.45 µm filter. The vector particles were either used directly for transduction or frozen at −80 °C.

### 4.4. Transduction

For transduction of NK-92 cells or peripheral blood NK cells, 2.5 × 10^5^ cells were seeded into a single well of a 24-well plate in 50 µL NK-92 or NK medium, respectively. We added 0.5 mL of the cell culture supernatant containing the lentiviral particles and 5 µL Vectofusin-1 (Miltenyi Biotec, Bergisch Gladbach, Germany) to the cells. Spinfection was performed at 400× *g* at 37 °C for 60 min. Afterwards, 0.5 mL of NK-92 or NK medium was added. Transgene expression was determined by flow cytometry 3 days post transduction. NK-92 cells were selected 3 days post transduction using 2 µg/mL puromycin for 2 weeks. Then, cells were used for experiments and aliquots were frozen at different time points. After thawing, cells were cultured for one week, before they were used for experiments. Peripheral blood NK cells were transduced 7 to 11 days after isolation.

### 4.5. Flow Cytometry

Flow cytometry analysis was performed using the MACSQuant X (Miltenyi Biotec, Bergisch Gladbach, Germany) and data were analyzed using MACSQuantify (Miltenyi Biotec, Bergisch Gladbach, Germany) and FlowJo10 (FlowJo, Ashland, OR, USA). CCR4 and CCR2B expression were determined by cell labeling with anti-CCR4 antibodies (see Table 1) in a 1:100 dilution and anti-CCR2 antibodies (see Table 1) in a 1:20 dilution, respectively. Cells were incubated with the antibodies for 10 min at RT, followed by two washing steps and final resuspension in 200 µL FACS buffer (PBS, 2% FCS, 1 mM EDTA). All other antibodies were diluted 1:200 and cells were stained for 10 min at 4 °C. Cells were washed twice and resuspended in 200 µL FACS buffer prior to analysis. NK cell purities after isolation were assessed with CD45, CD3, CD4, CD8, CD14, CD16, CD19 and CD56 staining (see Table 1 for antibodies). NK cells were identified as CD45^+^, CD3/14/19^−^, CD56^+^ and CD16^+/−^ cells. NK-92 cells were analyzed for CD56, NTB-A, NKG2D, 2B4, DNAM-1, NKp44, NKp46 and NKp30 expression (see Table 1 for antibodies). Primary cytokine-activated NK cells were analyzed for CD56, CD16, CD57, KIR2D, NKp44 and NKG2A (see Table 1 for antibodies). For the CellTrace™ analysis, cells were stained prior to transduction with CellTrace Violet (Thermo Fisher, Waltham, MA, USA) at a working concentration of 1 µM in 1 mL PBS for 20 min at 37 °C protected from light. Then, 5 mL of RPMI (10% FCS) was added and incubated for 5 min to remove remaining dye. Cells were centrifuged at 300× *g* for 5 min and resuspended in the appropriate volume of NK MACS medium.

### 4.6. qPCR

To quantify RNA expression, qPCR was performed using *β*-actin as reference gene. At first, RNA was isolated using the Quick-RNA MiniPrep kit according to the manufacturer’s instructions (Zymo Research, Irvine, CA, USA). DNaseI treatment was carried out as recommended. We diluted 1 µg of RNA in 16 µL H_2_O and 4 µL iScript Reverse Transcription Supermix (NEB) was added to synthesize the cDNA. The following primer pairs were used:

bActin_fw_qPCR (CACCATTGGCAATGAGCGGTTC), bActin_rev_qPCR (AGGTCTTTGCGGATGTCCACGT), CCL2_fw_qPCR (AGAATCACCAGCAGCAAGTGTCC), CCL2_rev_qPCR (TCCTGAACCCACTTCTGCTTGG), CCL22_fw_qPCR (TCCTGGGTTCAAGCGATTCTCC), CCL22_rev_qPCR (GTCAGGAGTTCAAGACCAGCCT), CCR2_fw_qPCR (CAGGTGACAGAGACTCTTGGGA), CCR2_VarB_rev_qPCR (CTTCTGAACTTCTCCCCAACGA), CCR4_fw_qPCR (CTCTGGCTTTTGTTCACTGCTGC), CCR4_rev_qPCR (AGCCCACAGTATTGGCAGAGCA) (Thermo Fisher, Waltham, MA, USA).

### 4.7. Migration Assay

A migration assay utilizing PET cell culture inserts with a 6.4 mm diameter and 5 µm pore size was used to assess the migration potential of NK cells. We loaded 2.5 × 10^5^ NK-92 (2 × 10^5^ NK) cells in the cell culture inserts at a total volume of 100 µL RPMI (without phenol red) (Thermo Fisher, Waltham, MA, USA). The inserts were placed in 24-well dishes containing 50 ng/mL of the respective chemokine (CCL22/CCL2 (Peprotech, Cranbury, NJ, USA)) diluted in 600 µL RPMI (without phenol red). After 2 h (NK-92) or 4 h (NK) of incubation at 37 °C and 5% CO_2_, cells and medium of the lower chamber were harvested and the number of migrated cells was quantified by flow cytometry. Primary NK cells were used 7 days post transduction.

To stimulate the migration towards cells expressing the cognate ligand, HEK293T cells were stably transduced with the chemokines CCL22 or CCL2, respectively. We seeded 2.5 × 10^4^ transduced HEK293T cells into a single well of a 24-well plate in 600 µL RPMI (without phenol red) (Thermo Fisher, Waltham, MA, USA). The next day, cell culture inserts containing 2.5 × 10^5^ cells in 100 µL RPMI (without phenol red) (Thermo Fisher, Waltham, MA, USA) were placed in each well. After 4 h of incubation at 37 °C, cell culture inserts were removed and the migrated NK-92 cells were stained with anti-CD56 antibody (1:600) and quantified by flow cytometry.

### 4.8. Cytotoxicity Assay

Transduced and non-transduced HEK293T cells were stained with CellTrace Violet (Thermo Fisher, Waltham, MA, USA) at a working concentration of 1 µM in 1 mL PBS for 20 min at 37 °C protected from light. Then, 5 mL of RPMI (10% FCS) was added and incubated for 5 min to remove remaining dye. Cells were centrifuged at 300× *g* for 5 min and resuspended in the appropriate volume of RPMI (without phenol red) to a final concentration of 4.2 × 10^4^ cells/mL. Then, 2.5 × 10^4^ transduced HEK293T cells were seeded into a single well of a 24-well plate in 600 µL RPMI (without phenol red). The next day, 33.3 × 10^4^ NK-92 cells in 100 µL RPMI (without phenol red) were added. After 20 h, cells were harvested by pipetting and propidium iodide (PI) was added as a dead cell marker in a 1:200 dilution. Afterwards, living cells were counted by flow cytometry (CellTrace Violet-positive; PI-negative). Specific lysis was calculated by diving each cell count with the count obtained in a control well containing HEK293T cells only.

### 4.9. Statistic Analysis

Statistical analyses were performed with GraphPad Prism 6 software (GraphPad Software, San Diego, CA, USA). Statistical significance was determined by applying the unpaired two-tailed *t*-test as indicated. For Figure 5, the paired two-tailed *t*-test was applied. *: *p* ≤ 0.05. **: *p* ≤ 0.01. ***: *p* ≤ 0.001. ****: *p* ≤ 0.0001.

## Figures and Tables

**Figure 1 ijms-24-03129-f001:**
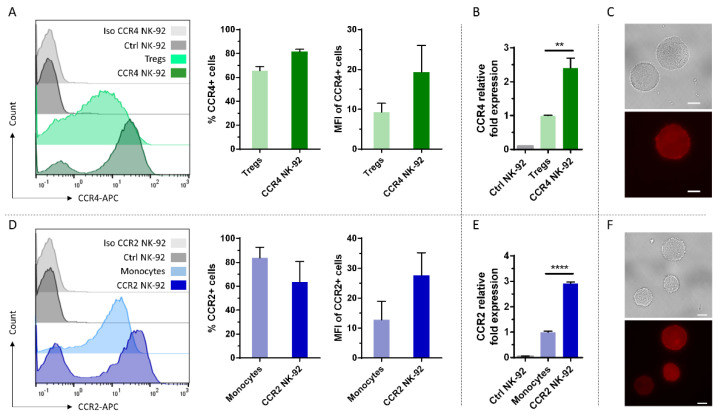
Efficient overexpression of CCR4 and CCR2B on NK-92 cells. (**A**,**D**) CCR4 or CCR2B expression was analyzed by flow cytometry on NK-92 cells transduced with CCR4- or CCR2B-encoding lentiviral particles and selected for 14 days using 2 µg/mL puromycin. T regs and monocytes that naturally express CCR4 and CCR2, respectively, were used for comparison. Histogram overlay shows the intensity of the APC signal of the CCR4/CCR2-specific antibody. Histogram shows one representative out of *n* = 3 independent experiments; bar chart shows CCR4-/CCR2-positive fractions and their MFIs. Mean values with SD are shown from *n* = 3 independent experiments. Ctrl NK-92: Untransduced NK-92 cells stained with CCR4 or CCR2 antibodies. (**B**,**E**) qPCR analysis of CCR4 or CCR2B transduced NK-92 cells versus T regs or monocytes, respectively. Ctrl NK-92: Untransduced NK-92 cells. Mean values with SD are shown from *n* = 3 independent experiments with three technical replicates each (**C**,**F**) Immunofluorescence microscopy of CCR4/CCR2B transduced NK-92 cells. Upper panel: bright field, lower panel: fluorescence staining of the respective antibodies directed against CCR4 and CCR2 chemokine receptors. Scalebar represents 10 µm. **: *p* ≤ 0.01. ****: *p* ≤ 0.0001.

**Figure 2 ijms-24-03129-f002:**
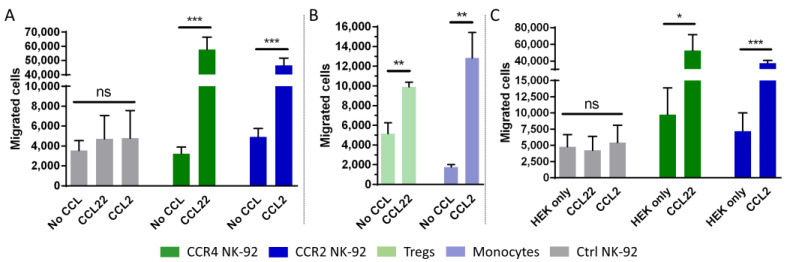
CCR4- and CCR2B-transduced NK-92 cells migrate towards CCL22 and CCL2. (**A**,**B**) NK-92 cells, T regs or monocytes were added to the cell culture insert and assessed for their potential to migrate to the lower chamber containing 50 ng/mL CCL22 or CCL2. The absolute number of migrated cells was counted by flow cytometry. Mean values with SD of *n* = 3 independent experiments with three technical replicates each are shown for NK-92 cells. Mean values with SD of one representative experiment out of *n* = 3 independent experiments with three technical replicates each are shown for T regs and monocytes. Ctrl NK-92: Untransduced NK-92 cells. No CCL: No chemokine added to the medium. (**C**) NK-92 cells with or without CCR4 and CCR2B were added to the cell culture inserts and assessed for their potential to migrate to the lower chamber in which untransduced (HEK only) or modified HEK293T cells stably expressing CCL22 or CCL2 (CCL22/CCL2) had been seeded the day before. Mean values with SD of *n* = 3 independent experiments with two technical replicates each are shown. The absolute number of migrated cells was determined by flow cytometry. *: *p* ≤ 0.05. **: *p* ≤ 0.01. ***: *p* ≤ 0.001. ns = not significant.

**Figure 3 ijms-24-03129-f003:**
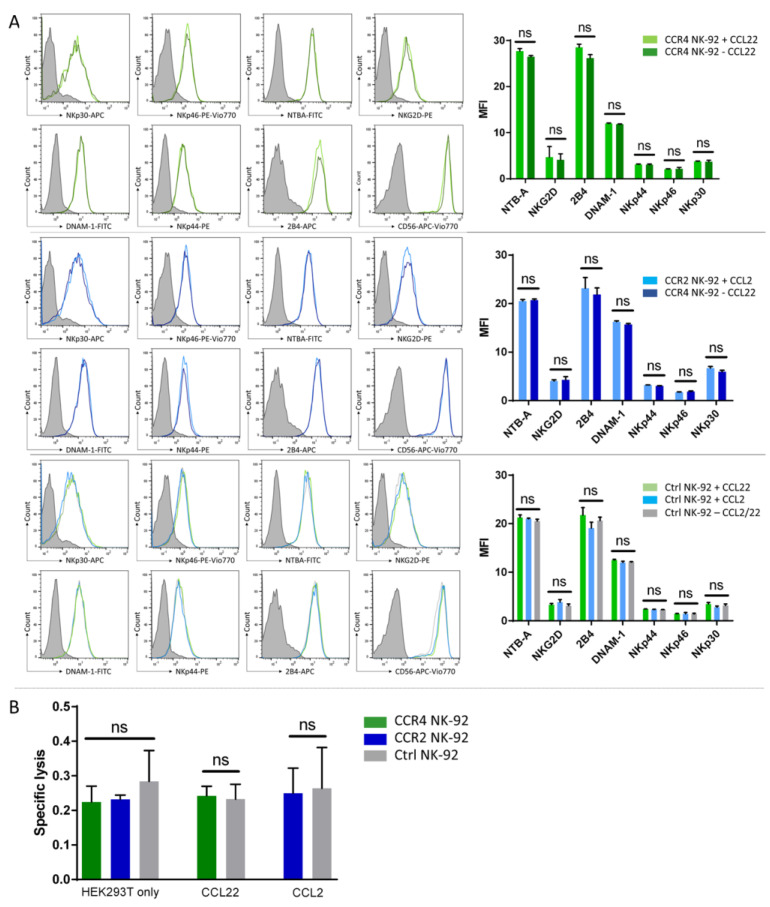
CCR4 or CCR2B expression does not alter the cytotoxic activity of NK-92 cells. (**A**) CCR4-/CCR2-NK-92 and control cells were incubated with the respective ligands (CCL22/CCL2) for 16 h. After that, expression of an array of activating NK cell receptors was analyzed by flow cytometry. Mean MFI values with SD of *n* = 3 independent experiments are shown. Histogram overlays show the intensity of each receptor’s antibody, the isotype staining is presented as gray shaded histogram. Histograms show one representative out of *n* = 3 independent experiments. (**B**) NK-92 cells were co-cultured directly with untransduced HEK293T (HEK293T only) and modified HEK293T cells stably overexpressing CCL22 or CCL2 chemokines (CCL22/CCL2). Specific lysis was assessed by flow cytometry. Mean values with SD of *n* = 3 independent experiments with two technical replicates each are shown for NK-92 cells. Ctrl NK-92: Untransduced NK-92 cells. ns = not significant.

**Figure 4 ijms-24-03129-f004:**
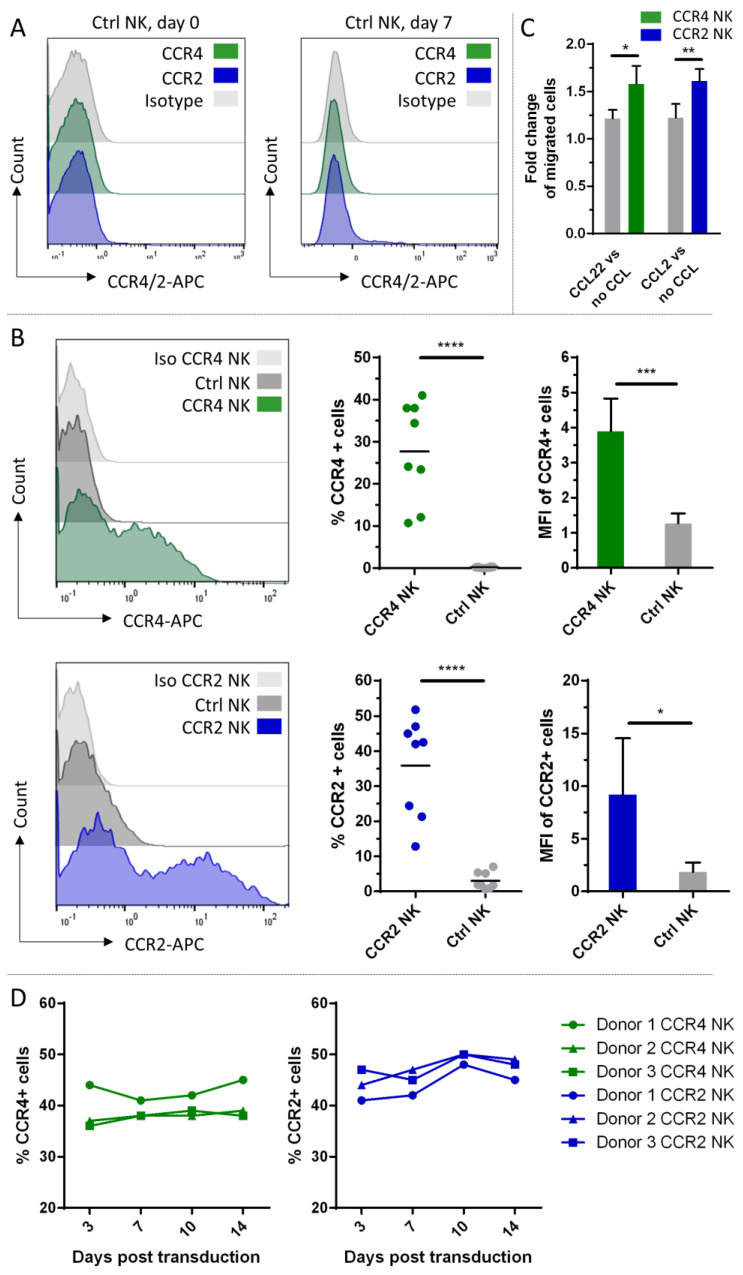
CCR4 or CCR2B expression on peripheral blood NK cells and functional testing of the genetically modified cells. (**A**) Freshly isolated NK cells from peripheral blood (day 0) and primary NK cells activated with IL-2 and IL-15 (day 7) were analyzed by flow cytometry for expression of CCR4 and CCR2B. (**B**) CCR4 or CCR2B expression was analyzed by flow cytometry on peripheral blood NK cells 7 days after transduction with CCR4- or CCR2B-encoding *γ*-retroviral particles. Histogram overlay shows the intensity of the APC signal of the CCR4/CCR2-specific antibody. Histograms show one representative out of *n* = 8 independent experiments; bar chart shows CCR4-/CCR2B-positive fractions (*n* = 8) and their MFIs (*n* = 5) of each donor. Mean values with SD are shown from *n* = 8 or *n* = 5 independent experiments. Ctrl NK: untransduced NK cells from the same donor stained with CCR4 or CCR2 antibody. (**C**) Seven days after transduction, NK cells were added to cell culture inserts and assessed for their potential to migrate to the lower chamber containing 50 ng/mL CCL22 or CCL2. The absolute number of migrated cells was counted by flow cytometry. Fold changes of migration against the respective ligand over migration against medium only with SD of *n* = 4 (CCR2) or *n* = 3 (CCR4) independent experiments with three technical replicates each are shown. Donors with less than 15% transduction efficiency were excluded from the migration assays. Ctrl NK: untransduced NK cells. no CCL: No chemokine added to the medium. (**D**) CCR4 or CCR2B expression was analyzed at different time points within 14 days after transduction. Graph shows frequencies of CCR4-/CCR2B-positive fractions (*n* = 3) at different time points. *: *p* ≤ 0.05. **: *p* ≤ 0.01. ***: *p* ≤ 0.001. ****: *p* ≤ 0.0001.

**Figure 5 ijms-24-03129-f005:**
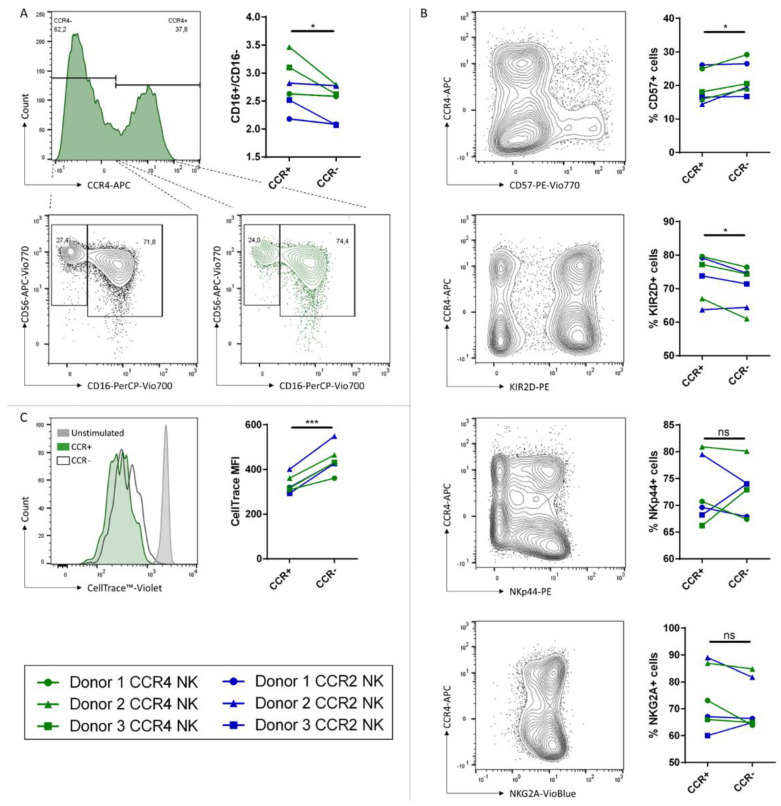
Characterization of different NK cell populations between transduced and untransduced NK cell fractions. Histogram plots show one representative out of *n* = 3 donors transduced with CCR4. Histograms from the CCR2B transductions were similar. Discrimination between transduced (CCR+) and untransduced (CCR-) fractions was performed by using the same gating strategy seen in histogram A. (**A**) CD56 and CD16 expression of transduced and untransduced fraction of a donor. Graph shows ratios of CD16-positive populations (CD16+) frequencies divided by CD16-negative populations (CD16−) frequencies. (**B**) Histograms show a representative distribution of cells by their expression of CCR4 and CD57, KIR2D, NKp44 or NKG2A. Graphs show frequencies of cells expressing CD57, KIR2D, NKp44 or NKG2A. (**C**) Cells were analyzed for proliferative potential via CellTrace™ Violet staining. Histogram shows one representative that was transduced with CCR4. Graph shows the MFIs of each CellTrace signal of *n* = 3 independent experiments on 3 donors. *: *p* ≤ 0.05. ***: *p* ≤ 0.001. ns = not significant.

**Table 1 ijms-24-03129-t001:** Antibody list.

Antibody	Conjugate	Manufacturer
2B4 (REA112)	APC	Miltenyi Biotec, Bergisch Gladbach, Germany
CCR2 (REA264)	APC	
CCR4 (REA279)	APC	
CD14 (REA599)	APC	
CD16 (REA423)	PerCP-Vio 700	
CD19 (REA675)	PE-Vio770	
CD3 (REA613)	VioBlue	
CD4 (REA623)	PE	
CD45 (REA747)	Vio-Green	
CD56 (REA196)	APC-Vio 770	
CD57 (REA769)	PE-Vio770	
CD8 (REA734)	FITC	
DNAM-1 (REA1040)	FITC	
Isotype Control (REA293)	APC	
Isotype Control (REA293)	FITC	
Isotype Control (REA293)	PE	
Isotype Control (REA293)	PE-Vio770	
Isotype Control (REA293)	APC-Vio770	
KIR2D (REA1042)	PE	
NKG2A (S19004C)	Pacific Blue	Biolegend, San Diego, CA, USA
NKG2D (REA1228)	PE	Miltenyi Biotec, Bergisch Gladbach, Germany
NKp30 (REA823)	APC	
NKp44 (REA1163)	PE	
NKp46 (REA808)	PE-Vio770	
NTB-A (REA339)	FITC	

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
