# Peer review of "Efficient Redirection of NK Cells by Genetic Modification with Chemokine Receptors CCR4 and CCR2B"

_ijms, 2023, doi:10.3390/ijms24043129_

Round 1
Reviewer 1 Report
In this study NK-92 and primary NK cells have been transduced to express CCR2 or CCR4B with the aim to increase their migration properties towards the TME which represents
a critical need in NK-based immunotherapies against solid tumors. The study has been well designed and performed and shows that NK-92 can be efficiently transduced. CCR2+/CCR4+ NK-92 can acquire specific migratory abilites without alterations in their phenotype and cytolytic function. The results obtained with primary NK cells are less brilliant as it can be expected when primary cells are transduced.
In view of a clinical translation the transduction efficacy on primary NK cells should be improved. However the study describes an interesting approach to profitably harness NK cells against solid tumors.
I have just a few observations that the authors could consider.
-I guess that the migration efficacy of transduced primary NK cells (fig 4C) is correlated to the frequency of CCR2+ or CCR4+ NK cells. This could be shown , it could corroborate the results.
-In primary NK cells both CCR4 and CCR2 show a heterogenous expression (fig 4b). I wonder whether the higher CCRX expression levels define a specific NK cell subset, such as highly proliferating, less mature and possibly less cytotoxic NK cells CD16-/DIM NKG2A+ KIR-CD57- etc or if transduced cells are homogenously distributed in the different NK cell subsets. This could have an impact on NK cells anti-tumor efficacy.
-In both transduction systems, NK-92 and primary NK cells, how long the CCRX surface expression is maintained?
minor
1-In the abstract at line 19 CCR4 an CCR2B are indicated in the wrong order, based on their expression indicated at line 20.
2-Fig 1D on the x-axis it’s CCR2 not CCR4
Author Response
Revised manuscript:
Efficient redirection of NK cells by genetic modification with chemokine receptors CCR4 and CCR2B
Response to Reviewer 1:
In this study NK-92 and primary NK cells have been transduced to express CCR2 or CCR4B with the aim to increase their migration properties towards the TME which represents
a critical need in NK-based immunotherapies against solid tumors. The study has been well designed and performed and shows that NK-92 can be efficiently transduced. CCR2+/CCR4+ NK-92 can acquire specific migratory abilites without alterations in their phenotype and cytolytic function. The results obtained with primary NK cells are less brilliant as it can be expected when primary cells are transduced.
In view of a clinical translation the transduction efficacy on primary NK cells should be improved. However the study describes an interesting approach to profitably harness NK cells against solid tumors.
We thank reviewer 1 for the critical assessment of our manuscript and below standing comments to improve our study.
I have just a few observations that the authors could consider.
-I guess that the migration efficacy of transduced primary NK cells (fig 4C) is correlated to the frequency of CCR2+ or CCR4+ NK cells. This could be shown , it could corroborate the results.
We saw significant migration for transduction efficiencies above 15 % as stated in the manuscript. However, it was hard to correlate absolute numbers of migrated cells as we observed also major backround migration effects between different donors. Certainly, also additional factors beside the chemokine receptor itself contributes to the efficient migration of cells. We tried to address this issue with the following sentence:
Migration experiments were carried out after each round of transduction, however only for transduction efficiencies greater than 15 %, significant migration behavior could be observed. (page 6/ line 210)
-In primary NK cells both CCR4 and CCR2 show a heterogenous expression (fig 4b). I wonder whether the higher CCRX expression levels define a specific NK cell subset, such as highly proliferating, less mature and possibly less cytotoxic NK cells CD16-/DIM NKG2A+ KIR-CD57- etc or if transduced cells are homogenously distributed in the different NK cell subsets. This could have an impact on NK cells anti-tumor efficacy.
We performed additional experiments on three donors to address this comment. Indeed, we observed that the transduced NK cells have predominantly a CD16+/CD57-/KIR+ and more proliferative phenotype. We added Figure 5 to show these results. We also added a section to the results as well as to the discussion part.
Results: page 6/ lines 219-238
Figure: page 8
Discussion: page 10/ lines 325-345
-In both transduction systems, NK-92 and primary NK cells, how long the CCRX surface expression is maintained?
For primary cells: We added information about the stability of transgene expression in primary NK cells to figure 4. (see Figure 4, panel D) We did not see any decrease within 14 days post transduction.
For NK-92 cells: We assessed the stability of transgene expression under puromycin selection. We added following sentence to the results section.
While transgene expression of CCR4 was constant over four weeks after selection, CCR2B expression was lost by 30 % of the cells during that time (data not shown). (page 3/ line 103-105)
minor
1-In the abstract at line 19 CCR4 an CCR2B are indicated in the wrong order, based on their expression indicated at line 20.
2-Fig 1D on the x-axis it’s CCR2 not CCR4
We thank you for seeing these mistakes. We adjusted the manuscript accordingly.
Thanks again for your comment. We think that your comments significantly improved our manuscript. We hope that we answered all your comments in a satisfactory manner.
Reviewer 2 Report
In this manuscript, Frederik Fabian Feigl et al. present a work on the NK cell genetic modification (both NK-92 cell line and primary NK cells) with the CCR4 and CCR2B chemokine receptors aimed to redirect these cells to migrate towards CCL22 or CCL2 produced in solid tumors. The study is of clear design. Background is described properly. The manuscript is well written. Authors present data of the effective lentiviral transduction of NK-92 cells and less effective but still significant retroviral transduction of primary cytokine-activated NK cells and show gene and surface expression of the CCR4 and CCR2B. The increased chemotactic activity of the genetically engineered cells has been shown in vitro. The authors are aware of the study limitations resulted from expected discrepancies in vitro and in vivo systems. Nevertheless, this work shows the possibility of the NK cell genetic modification with different chemokine receptors to influence the cell traffic. The choice of a receptor may depend on the intratumoral chemokine secretion pattern of a particular patient. In general, this approach may improve the cell-based cancer therapy.
There are no major concerns to the manuscript. Still, addressing several minor comments may help to improve the overall quality of the manuscript.
1. In figure 1 legend, three independent experiments are mentioned. Those experiments were three independent transductions or three measurements after one transduction? It should be clarified. Did the CCR4 and CCR2B expression levels vary over time?
2. How long the CCR4 and CCR2B expression levels in primary NK cells were stable after the transduction? In what day after the transduction the NK cells were used in functional assays? It should be clarified.
3. Regulatory T cells are designated as Tregs on page 1, however further in the text the term T reg is used. Either Treg or T reg should be chosen.
Author Response
Revised manuscript:
Efficient redirection of NK cells by genetic modification with chemokine receptors CCR4 and CCR2B
Response to Reviewer 2:
In this manuscript, Frederik Fabian Feigl et al. present a work on the NK cell genetic modification (both NK-92 cell line and primary NK cells) with the CCR4 and CCR2B chemokine receptors aimed to redirect these cells to migrate towards CCL22 or CCL2 produced in solid tumors. The study is of clear design. Background is described properly. The manuscript is well written. Authors present data of the effective lentiviral transduction of NK-92 cells and less effective but still significant retroviral transduction of primary cytokine-activated NK cells and show gene and surface expression of the CCR4 and CCR2B. The increased chemotactic activity of the genetically engineered cells has been shown in vitro. The authors are aware of the study limitations resulted from expected discrepancies in vitro and in vivo systems. Nevertheless, this work shows the possibility of the NK cell genetic modification with different chemokine receptors to influence the cell traffic. The choice of a receptor may depend on the intratumoral chemokine secretion pattern of a particular patient. In general, this approach may improve the cell-based cancer therapy.
We thank reviewer 2 for the critical assessment of our manuscript and below standing comments to improve our study.
There are no major concerns to the manuscript. Still, addressing several minor comments may help to improve the overall quality of the manuscript.
- In figure 1 legend, three independent experiments are mentioned. Those experiments were three independent transductions or three measurements after one transduction? It should be clarified. Did the CCR4 and CCR2B expression levels vary over time?
We performed three independent measurements at different time points after a single transduction and selection. We added this section to mat/meth: see below.
NK‑92 cells were selected 3 days post transduction using 2 µg/mL puromycin for 2 weeks. Then, cells were used for experiments and aliquots were frozen at different time points. After thawing, cells were cultured for one week, before they were used for experiments.(page 12/ lines 430-433)
We assessed the stability of transgene expression under puromycin selection. We added following sentence to the results section.
While transgene expression of CCR4 was constant over four weeks after selection, CCR2B expression was lost by 30 % of the cells during that time (data not shown). (page 3/ line 103-105)
- How long the CCR4 and CCR2B expression levels in primary NK cells were stable after the transduction? In what day after the transduction the NK cells were used in functional assays? It should be clarified.
We added information about the stability of transgene expression in primary NK cells to figure 4. (see Figure 4, panel D) We did not see any decrease within 14 days post transduction.
Cells were used in Boyden chamber experiments 7 days after transduction. We added this information into Figure legend 4C. and to Materials and methods section (page 13/ line 477-478)
Primary NK cells were used 7 days post transduction. (page 13/ line 477-478)
- Regulatory T cells are designated as Tregs on page 1, however further in the text the term T reg is used. Either Treg or T reg should be chosen.
It was changed to T reg throughout the manuscript
Additionally, we performed experiments on three donors to assess the phenotype of the transduced primary NK cells. We found out that the transduced NK cells have predominantly a CD16+/CD57-/KIR+ and more proliferative phenotype. We added Figure 5 to show these results. We also added a section to the results as well as to the discussion part.
Results: page 6/ lines 219-238
Figure: page 8
Discussion: page 10/ lines 325-345
Thanks again for your comment. We think that your comments significantly improved our manuscript. We hope that we answered all your comments in a satisfactory manner.